

# Spectrophotometric determination of L-α-glycerylphosphorylcholine in pharmaceutical formulations and industrial equipment cleaning rinse water with the WAKO Phospholipids C assay kit

Pavel Anatolyevich Nikolaychuk

Laboratory of Chemical Analysis, Quality Assurance Department, LLC "Velpharm", Kurgan, Russian Federation

## ABSTRACT

A simple spectrophotometric method for the determination of L-α-glycerylphosphorylcholine in pharmaceutical formulations and industrial equipment cleaning rinse water using the enzyme glycerophosphocholine phosphodiesterase and the WAKO Phospholipids C assay kit was proposed. The method is based on the enzymatic hydrolysis of α-GPC to choline by glycerophosphocholine phosphodiesterase, the reaction of choline with the components of the assay kit, and the colourimetric determination of the formed product. The calibration graph is linear in the range from 1 to 40 mg/l of α-GPC, the molar attenuation coefficient is 1,110 m$^2$/mol, the limit of detection is 1 mg/l, the limit of quantification is 3.3 mg/l, the method is selective with respect to the common excipients, shows a good accuracy (the relative uncertainty does not exceed 7%) and precision (the relative standard deviation does not exceed 5.5%), does not require lengthy sample preparation and sophisticated laboratory equipment and is suitable for the routine analysis of pharmaceutical formulations and industrial equipment cleaning rinse water.

## INTRODUCTION

L-α-glycerylphosphorylcholine (CAS registry number 28319-77-9, other names: (R)-2,3-dihydroxypropyl(2-(trimethylammonio)ethyl)phosphate, α-GPC, *sn*-glycero-3-phosphocholine, choline alfoscerate) is a water-soluble natural compound found in the human brain and a precursor to acetylcholine (*Choi, Hwang & Shin, 2020*; *Colucci et al., 2012*; *Lee, Young Choi & Won Suh, 2018*; *Moreno, 2003*; *Kidd, 2004*). It is used for the treatment of Alzheimer's disease (*Moreno, 2003*; *Hwang & Park, 2019*; *Selezneva, Kolykhalov & Gavrilova, 2020*; *Kim et al., 2017*; *Lanctôt et al., 2017*), other dementias and cognitive impairment (*Lee et al., 2017*; *Colucci et al., 2012*; *Doggrell & Evans, 2003*; *Sangiorgi et al., 1994*; *Putilina, 2020*), dry eye syndrome (*Choi, Hwang & Shin, 2020*), and as the

Corresponding author
Pavel Anatolyevich Nikolaychuk, npa@csu.ru

nootropic agent (*Colucci et al., 2012*; *Traini, Bramanti & Amenta, 2013*; *Tamura et al., 2021*; *Nobis & Husain, 2018*). It also could increase a physical performance (*Bellar, LeBlanc & Campbell, 2015*; *Marcus et al., 2017*; *Bogolepova et al., 2021*), could enhance the growth hormone secretion (*Kawamura et al., 2012*), and could be a possible cancer biomarker (*Jia et al., 2016*; *Moestue et al., 2012*; *Smith et al., 2017*). α-GPC is a non-prescription drug in many countries (*Kim & Cho, 2019*), and is manufactured in large quantities (*Van Hoogevest & Wendel, 2014*). *e.g.*, the Russian State Register of Pharmaceutical Products (https://grls.rosminzdrav.ru/Default.aspx) contains more than 30 different medications containing α-GPC including oral solutions, intravenous injections and capsules.

Cleaning of pharmaceutical equipment and determination of the product residues in the cleaning rinse water and on the manufacturing equipment surface is the important step in the pharmaceutical production (*Prabu & Suriyaprakash, 2010*; *Agalloco, 1992*; *Nassani, 2005*). Both European and US Pharmacopoeias do not contain monographs on α-GPC and do not propose methods for its assay, whereas the *State Pharmacopoeia of the Russian Federation (14th Edition) (2018)* proposes a method utilising the non-aqueous acid–base titration using crystal violet for end-point detection. However, this titration method is not suitable for quantification of microgram amounts of a substance in an aqueous solution (*Fritz, 1950*; *Riddick, 1958*). There are currently different analytical approaches available for the determination of various choline compounds (*Phillips, 2012*; *Wilson & Lorenz, 1979*), including NMR-spectrometry (*Holmes, Snodgrass & Iles, 2000*), liquid (*Zeisel et al., 2003*; *Andrieux et al., 2008*), gas (*Garavelli, 1972*) and ion-exchange chromatography (*Dorsey, Hansen & Gilbert, 1980*; *Laikhtman & Rohrer, 1999*), capillary electrophoresis (*Carter & Trenerry, 1996*), electrochemical methods (*Panfili et al., 2000*; *Pati et al., 2005*) and colourimetry. However, only a few of these methods were adopted specifically for the quantification of α-GPC. *Holmes, Snodgrass & Iles (2000)* developed a NMR-method for the determination of various choline compounds including α-GPC in milk after extraction. *Pomfret, Schurman & Zeisel (1989)* utilised preparative high-pressure liquid chromatography followed by gas chromatography with mass-spectrometric detection for analysis of different choline derivatives in human tissues. Later, *Holmes-McNary et al. (1996)* used the same method for determination of choline compounds in milk. Another liquid chromatographic method with mass-spectrometric detection for the determination of different choline compounds in tissues and foods was proposed by *Koc et al. (2002)*, and later modified by *Likes et al. (2007)*. Later, another modification of this method was proposed by *Zhao, Xiong & Curtis (2011)*. *Kozitsyna (2017)* described a liquid chromatographic determination of α-GPC in pharmaceutical formulations with refractometric detection. Another liquid chromatographic method with refractometric detection was proposed by *Zhao et al. (2020)*. *Gavrilin et al. (2012)* determined α-GPC in pharmaceutical formulations using capillary electrophoresis with indirect UV detection.

The analytical performance of these methods is compared in Table 1. As might be seen, the proposed methods utilising NMR, GC/MS and HPLC/MS are laborious, require time-consuming sample preparation and advanced instrumentation. The HPLC/refractometric and CE/UV methods are simple and quick, but lack both selectivity and sensitivity and are unsuitable for the determination of microgram amounts of α-GPC in cleaning rinse

water. The method of determination of trace amounts of pharmaceutical ingredients in the cleaning rinse water should be as rapid and simple as possible; therefore, spectrophotometric determination is a good choice. Although no spectrophotometric method designed especially for α-GPC was reported, there are four well-known groups of methods for free and total choline, and for various choline esters. The first group of methods is based on the precipitation of choline with ammonium diamminetetrakis (thiocyanato-N)chromate (Reinecke's salt), redissolution of the precipitate, and subsequent photometric determination of the coloured solution (*Kapfhammer & Bischoff, 1930*; *Beattie, 1936*; *Thornton & Broome, 1942*; *Engel, 1942*; *Glick, 1944*; *Marenzi & Cardini, 1943*; *Bandelin & Tuschhoff, 1951*). The second group of methods implements the precipitation of choline with potassium triiodide, and either the determination of liberated iodine (*Staněk, 1905*; *Sharpe, 1923*; *Hayashi, Unemoto & Miyaki, 1962*), or the redissolution and subsequent photometric determination of choline triiodide (*Appleton et al., 1953*). The third group of methods is based on the reaction of choline with phosphormolybdic acid (*Wheeldon & Collins, 1958*; *Wachsmuth & Van Koeckhoven, 1959*). The fourth group of methods uses enzymatic oxidation of choline by choline oxidase, and the subsequent determination of generated hydrogen peroxide (*Rahimi & Joseph, 2019*). This might be done by the reaction of hydrogen peroxide with phenol and 4-aminoantipyrine (*Woollard & Indyk, 1990*), with 3,5-dimethoxy-N-ethyl-N-(2-hydroxy-3-sulfopropyl)-sodium aniline and 4-aminoantipyrine (*Maeda et al., 1993*; *Mine, 1996*), with dichlorofluoscein (*Khan et al., 1992*), with 3,3′,5,5′-tetramethylbenzidine in presence of $MoS_2$ or $WS_2$ nanoparticles (*Nirala, Vinita & Prakash, 2018*; *Vinita, Nirala & Prakash, 2021*), with 2,2′-azino-bis(3-ethylbenzothiazoline-6-sulfonic acid (*Nikzad & Karami, 2018*), with $Fe^{2+}$ and o-phenylenediamine (*Chen et al., 2018*). In addition, choline might be estimated colorimetrically with triiodide/activated charcoal/molybdenum blue system (*Zimmerman & Ibrahim, 2018*), with *cis*-aconitic anhydride (*Böttcher, Pries & Van Gent, 1961*), or with 25,26,27,28-tetrahydroxycalix[4]arene-5,11,17,23-tetrasulfonic acid sodium salt (*Abd El-Rahman et al., 2019*).

The methods based on reactions with Reinecke's salt, potassium triiodide and phosphormolybdic acid require lengthy precipitation and redissolution steps and are not suitable for rapid routine analysis. The methods proposed by *Böttcher, Pries & Van Gent (1961)* and by *Abd El-Rahman et al. (2019)* are simple but employ rare and expensive reagents. On the other hand, enzymatic methods are rapid, simple, and many commercial assay kits utilising these methods are available. These commercial assay kits also contain phospholipases, which allows them to quantify not only the free choline, but also choline-containing phospholipids. The WAKO Phospholipids C assay kit manufactured by Wako Diagnostics (Mountain View, CA, USA) is based on the method of *Maeda et al. (1993)*. It contains phospholipase D, which hydrolyses phospholipids to choline, the choline then is oxidised by choline oxidase to betaine and hydrogen peroxide, which reacts with 3,5-dimethoxy-N-ethyl-N-(2-hydroxy-3-sulfopropyl)-sodium aniline and 4-aminoantipyrine in presence of peroxidase and produces a blue pigment that might be determined colourimetrically. α-GPC is not affected by phospholipase D, because it is usually produced from phospholipids by other enzymes (*Oyeneye et al., 2020*; *Hayashi*

**Table 1** The comparison of analytical performance of the proposed method with that of the other methods of α-GPC analysis available in the literature.

| Method | Reference | Range (mg/l) | Accuracy (%) | Precision (%) | Analysis time |
|---|---|---|---|---|---|
| NMR | *Holmes, Snodgrass & Iles (2000)* | 1.3–130 | 7 | Not specified | Sample preparation 1 h + spectrum acquisition 10-15 min |
| Preparative HPLC + GC/MS | *Pomfret, Schurman & Zeisel (1989)* | 0.5–5000 | 5 | 4 | Sample preparation 1 h + α-GPC retention time 15 min |
| HPLC/MS | *Koc et al. (2002)* and *Zhao, Xiong & Curtis (2011)* | 0.5–5000 | 12 | Not specified | α-GPC retention time 20 min |
| HPLC/Refractometric | *Kozitsyna (2017)* | 10,000 | Not specified | Not specified | α-GPC retention time 4 min |
| HPLC/Refractometric | *Zhao et al. (2020)* | 80–800 | 14 | 2 | α-GPC retention time 10 min |
| CE/UV | *Gavrilin et al. (2012)* | 125,000–375,000 | 2 | 1 | α-GPC retention time 15 min |
| Spectrophotometric | This work | 3–40 | 7 | 5.5 | Incubation time 10 min + absorbance measurement |

& Kornberg, 1954) like phospholipase A$_1$ (*Sonkar et al., 2019*; *Zhang, Liu & Wang, 2012*; *Bang, Kim & Kim, 2016*; *Liang et al., 2021*), phospholipase A$_2$ (*Blasi et al., 2006*; *Liang et al., 2021*), phospholipase B (*Kim et al., 2020*) and *Rhizopus chinensis* lipase (*Zhang, Wang & Liu, 2012*). However, the hydrolysis of α-GPC to choline might be achieved by the enzyme glycerophosphocholine phosphodiesterase (*Hayaishi & Kornberg, 1954*; *Sonkar et al., 2019*; *Oyeneye et al., 2020*). This way, by combining glycerophosphocholine phosphodiesterase with the reagents of the WAKO Phospholipids C assay kit, the quantitative colourimetric measurement of α-GPC becomes possible. The aim of this study is to develop a method of determination of α-GPC in pharmaceutical formulations and industrial equipment cleaning rinse water using the glycerophosphocholine phosphodiesterase and the WAKO Phospholipids C assay kit.

## MATERIALS & METHODS

### Reagents and equipment

The WAKO Phospholipids C assay kit was purchased from Wako Diagnostics (Mountain View, CA, USA). It consists of the colour reagent containing phospholipase D, choline oxidase, peroxidase, 3,5-dimethoxy-N-ethyl-N-(2-hydroxy-3-sulfopropyl)-sodium aniline and 4-aminoantipyrine and of the buffer solution, which composition and pH value are not specified. Native mold *sn*-glycerol-3-phosphocholine phosphodiesterase, in the form of the lyophilised powder containing tris(hydroxymethyl)aminomethane buffer salt, was purchased from Creative Enzymes (Shirley, NY, USA). The Hydranal moisture test kit for visual Karl-Fisher titration was purchased from Fluka (Buchs, Switzerland). Acetic anhydride (99%), methyl-4-hydroxybenzoate (99%) and propyl-4-hydroxybenzoate (99%) were purchased from Sigma-Aldrich (St. Louis, MO, USA). Mercury (II) acetate (analytical grade), potassium hydrogen phthalate (analytical grade), perchloric acid (analytical grade), crystal violet (analytical grade) and glycerol (analytical grade) were purchased from Khimreaktivsnab (Tashkent, Uzbekistan). Glacial acetic acid was purchased from Lenreaktiv (Saint Petersburg, Russia). Choline alfoscerate (98%) was purchased from Lipoid GmbH (Ludwigshafen am Rhein, Germany). Different pharmaceutical formulations containing α-GPC were purchased from Sotex (Moscow, Russia). The flat plates made of stainless steel 12X12H10T were used to model the cleaning of industrial equipment.

The analytical balance Sartorius Cubis MSA 225P-ICE-DI was used for weighting. The various micropipettes manufactured by Thermo Fisher Scientific (Waltham, MA, USA) were used for taking aliquots. The spectrophotometer Mettler Toledo UV7 was used for colorimetric measurements. The water bath Stegler WB-4 was used for sample incubations. The microburette Duran AS 5 ml was used for titration. The chemical glassware of the 2nd grade was used. Water for preparation of solutions was twice distillated and then deionised with Sartorius Arium Pro VF Ultrapure Water system.

### Preparation of the colour reagent
The contents of one vial with the colour reagent and one vial with the buffer from the WAKO Phospholipids C assay kit, and the contents of one vial with *sn*-glycerol-3-phosphocholine phosphodiesterase were mixed together. The vials were rinsed with the resulting solution; the rinses were collected into the 50 ml volumetric flask, and the volume of the solution was adjusted by water. The solution was stored in the refrigerator.

### Preparation of the 0.5% solution of crystal violet
A total of 0.50 g of crystal violet was weighted, dissolved in glacial acetic acid; the solution was transferred to the 100 ml volumetric flask, and the volume of the solution was adjusted by glacial acetic acid.

### Preparation of the 3% solution of mercury (II) acetate
A total of 3.00 g of mercury (II) acetate was weighted, dissolved in glacial acetic acid, the solution was transferred to the 100 ml volumetric flask, and the volume of the solution was adjusted by glacial acetic acid.

### Preparation and standardisation of the 0.1 M solution of perchloric acid
A total of 8.5 ml of perchloric acid was dissolved in ca. 900 ml of glacial acetic acid, then 30 ml of acetic anhydride was added, the solution was transferred to the 1,000 ml volumetric flask, and the volume of the solution was adjusted by glacial acetic acid. The water content in the solution was determined by visual Karl-Fisher titration using the Hydranal moisture test kit. If the water content was less than 0.1%, more water was added, and if it was greater than 0.2%, more acetic anhydride was added, and the water content determination was repeated. The solution was allowed to stand for 24 h. To standardise the solution 0.350 g of potassium hydrogen phthalate was weighted, dissolved in glacial acetic acid protected from light, and titrated with the perchloric acid solution using 0.05 ml of the 0.5% solution of crystal violet as indicator. This preparation procedure complies with *State Pharmacopoeia of the Russian Federation (14th Edition) (2018)*.

### Preparation of the 100 mg/l stock solution of α-GPC
A total of 1.0000 g of α-GPC was weighted, dissolved in glacial acetic acid, the solution was transferred to the 100 ml volumetric flask and the volume of the solution was adjusted by glacial acetic acid. The exact concentration of the solution was determined by titration. For this the aliquot of 5.0 ml of the prepared solution was transferred to the titration flask, 40 ml of acetic anhydride and 10 ml of the 3% solution of mercury (II) acetate was added, the solution was mixed and titrated with the standardised solution of 0.1 M perchloric

acid using 0.05 ml of the 0.5% solution of crystal violet as indicator. Then the appropriate aliquot of the prepared solution with the determined concentration was taken, transferred to the 1,000 ml volumetric flask, and the volume of the solution was adjusted by water. The stock solution was stored in a refrigerator.

### Preparation of working solutions of α-GPC

The working solutions of α-GPC with different concentrations ranging from 1 to 100 mg/l were prepared by appropriate dilution of the stock solution with water. The working solutions were prepared daily.

### Preparation of sample solutions from injections

The solutions for intravenous injections available on the Russian local market contain a 250 g/l solution of α-GPC. The contents of ten ampoules from the single package were collected into a beaker; the aliquot of 5.0 ml was taken, transferred to the 500 ml volumetric flask, dissolved in water, and the volume of the solution was adjusted by water. The aliquot of 5.0 ml of the prepared solution was taken, transferred to another 500 ml volumetric flask, and the volume of the solution was adjusted by water. The concentration of α-GPC in the resulting solution equals 25 mg/l.

### Preparation of sample solutions from oral solutions

The oral solutions available on the Russian local market contain a 120 g/l solution of α-GPC. The contents of ten ampoules from the single package were collected into a beaker; the aliquot of 5.0 ml was taken, transferred to the 500 ml volumetric flask, dissolved in water, and the volume of the solution was adjusted by water. The aliquot of 5.0 ml of the prepared solution was taken, transferred to another 500 ml volumetric flask, and the volume of the solution was adjusted by water. The concentration of α-GPC in the resulting solution equals 12 mg/l.

### Preparation of sample solutions from capsules

The capsules available on the Russian local market contain 400 mg of α-GPC. The contents of ten capsules from the single package were collected into a beaker and dissolved in water; the solution was transferred to the 1,000 ml volumetric flask, dissolved in water, and the volume of the solution was adjusted by water. The aliquot of 5.0 ml of the prepared solution was taken, transferred to the 500 ml volumetric flask, and the volume of the solution was adjusted by water. The concentration of α-GPC in the resulting solution equals 40 mg/l.

### Preparation of model rinse water samples from injections

The contents of ten ampoules from the single package were collected into a beaker; the aliquot of 5.0 ml was taken, transferred to the 500 ml volumetric flask, dissolved in water, and the volume of the solution was adjusted by water. The aliquot of 1.0 ml of the prepared solution was taken, placed onto the flat plate made of stainless steel 12X12H10T, and allowed to dry in the fume hood. The plate was rinsed several times with water, the combined rinses were transferred to the 100 ml volumetric flask, and the volume of the solution was adjusted by water. The expected concentration of α-GPC in the model rinse water sample equals 25 mg/l.

### Preparation of model rinse water samples from oral solutions

The contents of ten ampoules from the single package were collected into a beaker; the aliquot of 5.0 ml was taken, transferred to the 500 ml volumetric flask, dissolved in water, and the volume of the solution was adjusted by water. The aliquot of 1.0 ml of the prepared solution was taken, placed onto the flat plate made of stainless steel 12X12H10T, and allowed to dry in the fume hood. The plate was rinsed several times with water, the combined rinses were transferred to the 100 ml volumetric flask, and the volume of the solution was adjusted by water. The expected concentration of α-GPC in the model rinse water sample equals 12 mg/l.

### Preparation of model rinse water samples from capsules

The contents of ten capsules from the single package were collected into a beaker; the aliquot of 5.0 ml was taken, transferred to the 500 ml volumetric flask, dissolved in water, and the volume of the solution was adjusted by water. The aliquot of 1.0 ml of the prepared solution was taken, placed onto the flat plate made of stainless steel 12X12H10T, and allowed to dry in the fume hood. The plate was rinsed several times with water, the combined rinses were transferred to the 100 ml volumetric flask, and the volume of the solution was adjusted by water. The expected concentration of α-GPC in the model rinse water sample equals 40 mg/l.

### General procedure for the determination of α-GPC

A total of 3.0 ml of the colour reagent was mixed with 1.0 ml of working or sample solution of α-GPC in a test tube. The blank solution was prepared by mixing 3.0 ml of the colour reagent with 1.0 ml of water in another test tube. The contents of the test tubes were mixed, placed in the water bath and incubated at the temperature of 37 °C for 10 min. Then the absorbance of the working or sample solution of α-GPC at the wavelength of 595 nm in the glass cuvette with the optical path length one cm was measured against the blank solution.

## RESULTS

### Selection of the wavelength

The working solutions of α-GPC with the concentration of 30 mg/l and the blank solution were prepared and treated as described in the general procedure, and the spectrum of the working solution was recorded in the wavelength interval from 190 to 1,100 nm with the wavelength step of 0.2 nm in the glass cuvette with the optical path length of one cm against the blank solution. The spectrum is shown in Fig. 1. The maximum absorbance was observed at the wavelength of 595 nm. This wavelength was chosen for all further measurements.

### Selection of the sample volume

The working solutions of α-GPC with the concentration of 30 mg/l and the blank solution were prepared as described in the general procedure. In the series of the test tubes, 3.0 ml of the colour reagent were mixed with the various volumes of the working solution ranging from 0.2 to 2.0 ml. The corresponding blank solutions were prepared in another series

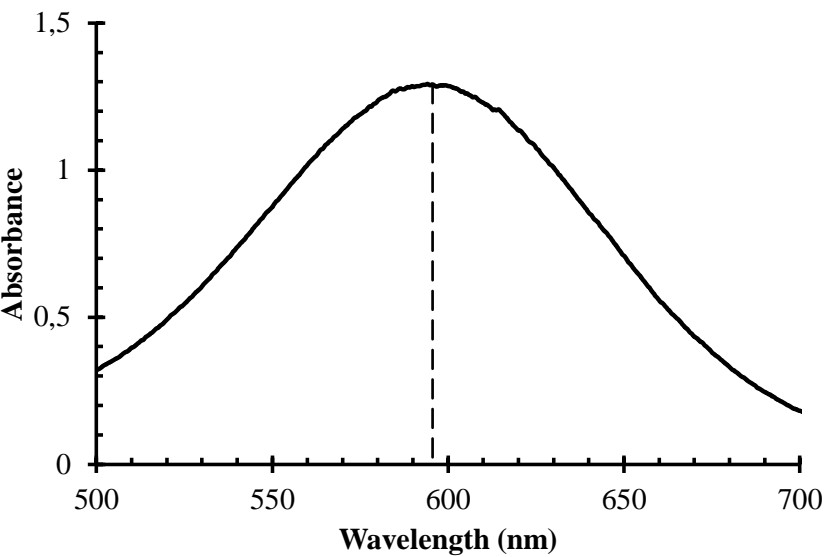

**Figure 1** **The visible spectrum of the coloured product of the enzymatic oxidation of α-GPC.** A solid curve represents the visible spectrum of the coloured product of the working solutions of α-GPC with the concentration of 30 mg/l that was recorded in the wavelength interval from 190 to 1,100 nm with the wavelength step of 0.2 nm in the glass cuvette with the optical path length of one cm against the blank solution. A dashed line outlines the maximum absorbance wavelength of 595 nm.

**Table 2** **The dependence of the absorbance of the coloured product on the sample volume.**

| Sample volume (ml) | Absorbance at 595 nm |
| --- | --- |
| 0.2 | 0.265 |
| 0.4 | 0.509 |
| 0.6 | 0.693 |
| 0.8 | 0.971 |
| 1.0 | 1.276 |
| 1.5 | 1.141 |
| 2.0 | 1.016 |

of test tubes by mixing 3.0 ml of the colour reagent with the various volumes of water. The solutions were incubated in the water bath at 37 °C for 10 min. The absorbances of prepared solutions with the different sample volume at the wavelength of 595 nm in the glass cuvette with the optical path length one cm were measured against the corresponding blank solutions. The results are shown in Table 2 and in Fig. 2. With the increase of the sample volume the molar concentration of α-GPC also increases, which favours the reaction rate to increase, but at the same point the molar concentrations of assay kit reagents and enzymes decrease by dilution, which favours the reaction rate to decrease. Therefore, at some point the reaction rate and the absorbance of the coloured product reach their maxima. The maximum absorbance was observed at the sample volume of 1 ml. This sample volume was chosen for all further measurements.

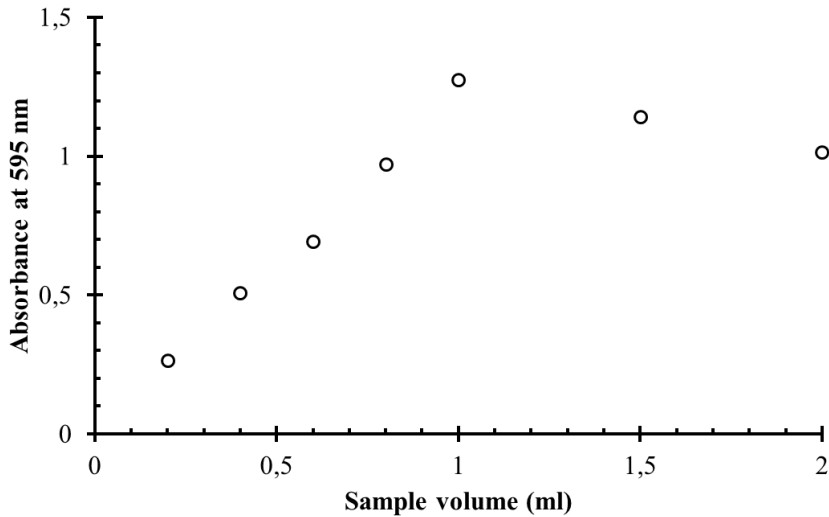

**Figure 2** **The dependence of the absorbance of the coloured reaction product on the sample volume.**

## Selection of the incubation time

The working solutions of α-GPC with the concentration of 30 mg/l and the blank solution were prepared as described in the general procedure. In the series of the test tubes, 3.0 ml of the colour reagent were mixed with 1.0 ml of working solution. The blank solution was prepared by mixing 3.0 ml of the colour reagent with 1.0 ml of water in another test tube. The solutions were incubated in the water bath at 37 °C for the various time intervals ranging from 1 to 20 min. The absorbances of prepared solutions with the different incubation time at the wavelength of 595 nm in the glass cuvette with the optical path length one cm were measured against the blank solution. The results are shown in Table 3 and in Fig. 3. As one might see from Fig. 3, after 10 min the overall reaction becomes very slow, and the further absorbance increase becomes asymptotic, which makes it pointless to wait for longer. In contrast, taking a lesser period of time decreases the absorbance significantly and this reduces the sensitivity. The optimal absorbance was observed at the incubation time of 10 min. This incubation time was chosen for all further measurements.

## Construction of the calibration curve

The working solutions of α-GPC with different concentrations ranging from 1 to 80 mg/l and the blank solution were prepared as described in the general procedure. In the series of the test tubes, 3.0 ml of the colour reagent were mixed with 1.0 ml of stock solution and prepared working solutions. The blank solution was prepared by mixing 3.0 ml of the colour reagent with 1.0 ml of water in another test tube. The solutions were incubated in the water bath at 37 °C for 10 min. The absorbances of prepared solutions with the different concentration of α-GPC at the wavelength of 595 nm in the glass cuvette with the optical path length one cm were measured against the blank solution. The results are shown in Table 4 and Fig. 4.

**Table 3** The dependence of the absorbance of the coloured product on the incubation time.

| Incubation time (min) | Absorbance at 595 nm |
| --- | --- |
| 1.0 | 0.263 |
| 2.0 | 0.481 |
| 3.0 | 0.656 |
| 4.0 | 0.803 |
| 6.0 | 1.025 |
| 8.0 | 1.176 |
| 10.0 | 1.282 |
| 15.0 | 1.355 |
| 20.0 | 1.406 |

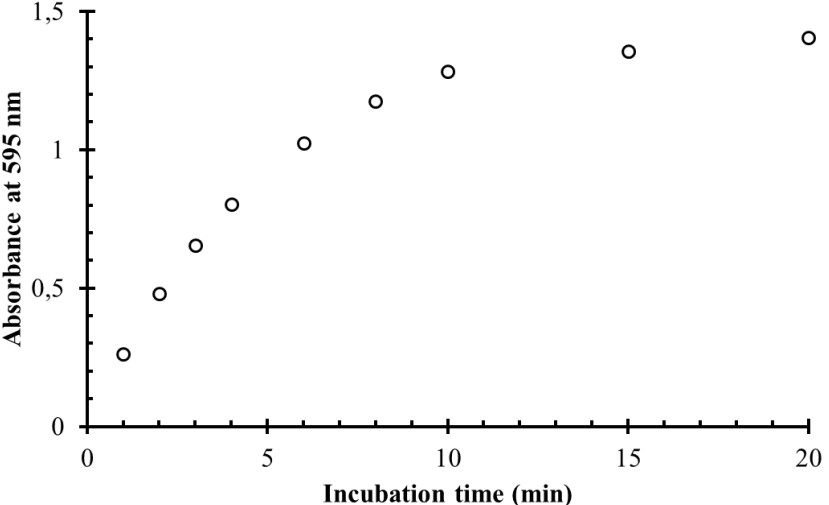

**Figure 3** The dependence of the absorbance of the coloured reaction product on the incubation time.

## Analytical performance

The analytical performance of the method was determined in accordance with *State Pharmacopoeia of the Russian Federation (14th Edition) (2018)* guidelines. The method was tested for linearity, limits of detection and quantification, selectivity, accuracy, and inter- and intra-day precision.

## Linearity

According to Fig. 2, the dependence of the absorbance of the coloured product at 595 nm on the concentration of α-GPC is linear in the range from 1 to 40 mg/l. The regression analysis was performed using the least-squares technique (*Adrain, 1808*). Additionally, the Ringbom's optimum range (*Ringbom, 1938*; *Ayres, 1949*; *Youmans & Brown, 1976*), the molar attenuation coefficient, and the Sandell's sensitivity coefficient (*Sandell, 1944*) were calculated. The parameters of the regression equation are listed in Table 5.

**Table 4  The dependence of the absorbance of the coloured product on the concentration of α-GPC.**

| Concentration of α-GPC (mg/l) | Absorbance at 595 nm |
|---|---|
| 1 | 0.029 |
| 2 | 0.061 |
| 4 | 0.158 |
| 5 | 0.221 |
| 10 | 0.447 |
| 15 | 0.618 |
| 20 | 0.844 |
| 30 | 1.288 |
| 40 | 1.710 |
| 60 | 2.447 |
| 80 | 2.790 |
| 100 | 2.931 |

**Figure 4  The dependence of the absorbance of the coloured reaction product on the concentration of α-GPC.** Empty circles represent the data points, and a dashed line corresponds to the regression equation.

## Limit of detection and limit of quantification

The limit of detection and the limit of quantification of the method were calculated according to *Currie (1999)*, *Shrivastava & Gupta (2011)* and *Little (2015)*. The values are presented in Table 5.

## Selectivity with respect to common excipients

According to the Russian State Register of Pharmaceutical Products (https://grls.rosminzdrav.ru/Default.aspx), intravenous injections contain only a solution of α-GPC

**Table 5  The parameters of the linear regression of the dependence of the absorbance of the coloured product on the concentration of α-GPC, and the analytical parameters of the method.**

| Parameter | Value |
|---|---|
| Slope and its confidence interval ($f = 7$, $p = 95\%$) (l/mg) | $0.0431 \pm 0.0004$ |
| Intercept and its confidence interval ($f = 7$, $p = 95\%$) | $-0.010 \pm 0.008$ |
| $R^2$ value | 0.9994 |
| Linearity range (mg/l) | 1–40 |
| Ringbom's optimum range (mg/l) | 5–16 |
| Molar attenuation coefficient and its confidence interval ($f = 7$, $p = 95\%$) (m²/mol) | $1{,}110 \pm 10$ |
| Sandell's sensitivity coefficient and its confidence interval ($f = 7$, $p = 95\%$) ($\mu$g/cm²) | $0.023 \pm 0.009$ |
| Limit of detection (mg/l) | 1.0 |
| Limit of quantification (mg/l) | 3.3 |

without excipients, oral solutions contain methyl-4-hydroxybenzoate and propyl-4-hydroxybenzoate, and capsules contain glycerol as the common excipients. The possible interference of these excipients as well as of acetic acid (because it was used in the stock solution preparation) was studied. For that, the 50 mg/l solutions of methyl-4-hydroxybenzoate and propyl-4-hydroxybenzoate, and the 1% solutions of glycerol and acetic acid were prepared. 3.0 ml of each solution were placed in the test tubes, 1.0 ml of the colour reagent was added to each one, and the solutions were incubated in the water bath at the temperature 37 °C for 60 min. No colour development was observed; this indicates that the tested excipients did not interfere.

However, the implementation of the WAKO Phospholipids C assay kit for the analysis of α-GPC in complex matrices is not possible. Blood and other bodily fluids contain phospholipids, which are also affected by this kit, and in this case it quantifies the phospholipids instead of α-GPC. Because one of the reaction steps includes the formation of hydrogen peroxide, any other enzymes and substrates that produce $H_2O_2$ also interfere. This implies that this method is not suitable for the determination of α-GPC in food and plant material, because many of raw natural ingredients contain both glucose and glucose oxidase, which lead to the $H_2O_2$ production.

## Accuracy

Four series of experiments were conducted. In the first series ten working solutions with the concentration of α-GPC equal to 5 mg/l, in the second series ten sample solutions from injections with the concentration of α-GPC equal to 25 mg/l, in the third series ten sample solutions from oral solution with the concentration of α-GPC equal to 12 mg/l, and in the fourth series ten sample solutions from capsules with the concentration of α-GPC equal to 40 mg/l were prepared. The solutions were treated as described in the general procedure, and then the absorbance of the coloured product was recorded; the concentrations of the solutions were calculated according to the regression equation, and the relative uncertainties were determined. The results are collected in Table 6.

**Table 6  The accuracy test of the method.**

| | Sample number | Absorbance at 595 nm | Concentration of $\alpha$-GPC (mg/l) | Relative uncertainty, % |
|---|---|---|---|---|
| | 1 | 0.198 | 4.83 | 3.47 |
| | 2 | 0.205 | 4.98 | 0.46 |
| | 3 | 0.213 | 5.16 | 3.27 |
| | 4 | 0.196 | 4.78 | 4.41 |
| Experiment 1: Working solution, 5 mg/l | 5 | 0.205 | 4.99 | 0.20 |
| | 6 | 0.215 | 5.22 | 4.46 |
| | 7 | 0.209 | 5.08 | 1.50 |
| | 8 | 0.207 | 5.03 | 0.63 |
| | 9 | 0.210 | 5.10 | 2.07 |
| | 10 | 0.203 | 4.94 | 1.25 |
| | Mean value | 0.207 | 5.01 | 2.17 |
| | 1 | 1.077 | 25.23 | 0.93 |
| | 2 | 1.076 | 25.21 | 0.83 |
| | 3 | 1.069 | 25.04 | 0.17 |
| | 4 | 1.059 | 24.81 | 0.76 |
| Experiment 2: Solution from intravenous injection, 25 mg/l | 5 | 1.060 | 24.82 | 0.73 |
| | 6 | 1.061 | 24.85 | 0.59 |
| | 7 | 1.063 | 24.89 | 0.44 |
| | 8 | 1.077 | 25.21 | 0.84 |
| | 9 | 1.062 | 24.88 | 0.50 |
| | 10 | 1.067 | 25.00 | 0.01 |
| | Mean value | 1.067 | 24.99 | 0.58 |
| | 1 | 0.501 | 11.85 | 1.26 |
| | 2 | 0.500 | 11.84 | 1.37 |
| | 3 | 0.503 | 11.90 | 0.81 |
| | 4 | 0.515 | 12.17 | 1.43 |
| Experiment 3: Solution from oral solution, 12 mg/l | 5 | 0.498 | 11.79 | 1.79 |
| | 6 | 0.502 | 11.89 | 0.95 |
| | 7 | 0.512 | 12.11 | 0.92 |
| | 8 | 0.498 | 11.79 | 1.75 |
| | 9 | 0.498 | 11.80 | 1.69 |
| | 10 | 0.498 | 11.78 | 1.85 |
| | Mean value | 0.502 | 11.89 | 1.38 |

**Table 6** (*continued*)

| | Sample number | Absorbance at 595 nm | Concentration of α-GPC (mg/l) | Relative uncertainty, % |
|---|---|---|---|---|
| | 1 | 1.704 | 39.77 | 0.57 |
| | 2 | 1.710 | 39.90 | 0.25 |
| | 3 | 1.723 | 40.20 | 0.50 |
| | 4 | 1.709 | 39.88 | 0.29 |
| Experiment 4: Solution from capsules, 40 mg/l | 5 | 1.718 | 40.10 | 0.24 |
| | 6 | 1.716 | 40.05 | 0.11 |
| | 7 | 1.708 | 39.86 | 0.35 |
| | 8 | 1.719 | 40.12 | 0.30 |
| | 9 | 1.722 | 40.20 | 0.49 |
| | 10 | 1.712 | 39.95 | 0.13 |
| | Mean value | 1.714 | 40.00 | 0.32 |

## Intra-day precision

Four series of experiments were conducted. In the first series ten working solutions with the concentration of α-GPC equal to 5 mg/l, in the second series ten sample solutions from injections with the concentration of α-GPC equal to 25 mg/l, in the third series ten sample solutions from oral solution with the concentration of α-GPC equal to 12 mg/l, and in the fourth series ten sample solutions from capsules with the concentration of α-GPC equal to 40 mg/l were prepared. The solutions were treated as described in the general procedure, and then the absorbance of the coloured product was recorded; the concentrations of the solutions were calculated according to the regression equation, and the relative standard deviations were determined. The results are collected in Table 7.

## Inter-day precision

Four series of experiments were conducted. In the first series a working solution with the concentration of α-GPC equal to 5 mg/l, in the second series a sample solution from injections with the concentration of α-GPC equal to 25 mg/l, in the third series a sample solution from oral solution with the concentration of α-GPC equal to 12 mg/l, and in the fourth series a sample solution from capsules with the concentration of α-GPC equal to 40 mg/l were prepared each day during consecutive five days. The solutions were treated as described in the general procedure, and then the absorbance of the coloured product was recorded; the concentrations of the solutions were calculated according to the regression equation, and the relative standard deviations were determined. The results are collected in Table 8.

## Accuracy for the determination of model rinse water samples

Three series of experiments were conducted. In the first series five model rinse water samples from injections with the concentration of α-GPC equal to 25 mg/l, in the second series five model rinse water solutions from oral solution with the concentration of α-GPC equal to 12 mg/l, and in the third series five model rinse water solutions from capsules with the concentration of α-GPC equal to 40 mg/l were prepared. The solutions were treated as described in the general procedure, and then the absorbance of the coloured

**Table 7  The intra-day precision test of the method.**

| Sample number | Experiment 1: Working solution, 5 mg/l | | Experiment 2: Solution from intravenous injection, 25 mg/l | | Experiment 3: Solution from oral solution, 12 mg/l | | Experiment 4: Solution from capsules, 40 mg/l | |
|---|---|---|---|---|---|---|---|---|
| | Absorbance at 595 nm | Concentration of $\alpha$-GPC (mg/l) | Absorbance at 595 nm | Concentration of $\alpha$-GPC (mg/l) | Absorbance at 595 nm | Concentration of $\alpha$-GPC (mg/l) | Absorbance at 595 nm | Concentration of $\alpha$-GPC (mg/l) |
| 1 | 0.208 | 5.06 | 1.073 | 25.12 | 0.498 | 11.79 | 1.714 | 40.00 |
| 2 | 0.210 | 5.11 | 1.059 | 24.81 | 0.513 | 12.13 | 1.707 | 39.84 |
| 3 | 0.213 | 5.18 | 1.073 | 25.12 | 0.517 | 12.22 | 1.719 | 40.13 |
| 4 | 0.202 | 4.93 | 1.058 | 24.79 | 0.516 | 12.21 | 1.708 | 39.86 |
| 5 | 0.211 | 5.13 | 1.058 | 24.77 | 0.498 | 11.78 | 1.721 | 40.16 |
| 6 | 0.203 | 4.94 | 1.060 | 24.83 | 0.514 | 12.17 | 1.721 | 40.15 |
| 7 | 0.210 | 5.10 | 1.075 | 25.18 | 0.504 | 11.94 | 1.704 | 39.78 |
| 8 | 0.207 | 5.05 | 1.058 | 24.79 | 0.509 | 12.04 | 1.707 | 39.84 |
| 9 | 0.200 | 4.86 | 1.073 | 25.12 | 0.511 | 12.09 | 1.713 | 39.97 |
| 10 | 0.214 | 5.21 | 1.058 | 24.79 | 0.499 | 11.81 | 1.716 | 40.04 |
| Mean value | 0.208 | 5.06 | 1.065 | 24.93 | 0.508 | 12.02 | 1.713 | 39.98 |
| SD | 0.005 | 0.114 | 0.008 | 0.177 | 0.008 | 0.176 | 0.006 | 0.141 |
| RSD (%) | 2.36 | 2.25 | 0.72 | 0.71 | 1.49 | 1.46 | 0.36 | 0.35 |

**Table 8  The inter-day precision test of the method.**

| Sample number | Experiment 1: Working solution, 5 mg/l | | Experiment 2: Solution from intravenous injection, 25 mg/l | | Experiment 3: Solution from oral solution, 12 mg/l | | Experiment 4: Solution from capsules, 40 mg/l | |
|---|---|---|---|---|---|---|---|---|
| | Absorbance at 595 nm | Concentration of α-GPC (mg/l) | Absorbance at 595 nm | Concentration of α-GPC (mg/l) | Absorbance at 595 nm | Concentration of α-GPC (mg/l) | Absorbance at 595 nm | Concentration of α-GPC (mg/l) |
| 1 | 0.205 | 4.99 | 1.058 | 24.78 | 0.514 | 12.15 | 1.717 | 40.06 |
| 2 | 0.197 | 4.80 | 1.059 | 24.81 | 0.502 | 11.89 | 1.705 | 39.80 |
| 3 | 0.214 | 5.20 | 1.078 | 25.23 | 0.530 | 12.53 | 1.730 | 40.37 |
| 4 | 0.202 | 4.91 | 1.058 | 24.78 | 0.505 | 11.95 | 1.708 | 39.85 |
| 5 | 0.210 | 5.09 | 1.091 | 25.55 | 0.501 | 11.85 | 1.736 | 40.52 |
| Mean value | 0.205 | 4.99 | 1.069 | 25.03 | 0.510 | 12.07 | 1.719 | 40.12 |
| SD | 0.007 | 0.155 | 0.015 | 0.348 | 0.012 | 0.279 | 0.014 | 0.317 |
| RSD (%) | 3.25 | 3.10 | 1.40 | 1.39 | 2.36 | 2.31 | 0.79 | 0.79 |

**Table 9  The accuracy test for the model rinse water solutions.**

| | Sample number | Absorbance at 595 nm | Concentration of α-GPC (mg/l) | Relative uncertainty, % |
|---|---|---|---|---|
| Experiment 1: Solution from intravenous injection, 25 mg/l | 1 | 1.007 | 23.60 | 5.60 |
| | 2 | 0.995 | 23.31 | 6.75 |
| | 3 | 0.996 | 23.33 | 6.67 |
| | 4 | 1.003 | 23.51 | 5.95 |
| | 5 | 1.000 | 23.43 | 6.29 |
| | Mean value | 1.000 | 23.44 | 6.25 |
| Experiment 2: Solution from oral solution, 12 mg/l | 1 | 0.481 | 11.40 | 4.98 |
| | 2 | 0.483 | 11.43 | 4.71 |
| | 3 | 0.474 | 11.24 | 6.37 |
| | 4 | 0.475 | 11.26 | 6.14 |
| | 5 | 0.472 | 11.19 | 6.76 |
| | Mean value | 0.477 | 11.30 | 5.79 |
| Experiment 3: Solution from capsules, 40 mg/l | 1 | 1.611 | 37.60 | 5.99 |
| | 2 | 1.607 | 37.52 | 6.19 |
| | 3 | 1.609 | 37.56 | 6.11 |
| | 4 | 1.607 | 37.52 | 6.19 |
| | 5 | 1.614 | 37.67 | 5.81 |
| | Mean value | 1.610 | 37.58 | 6.06 |

product was recorded; the concentrations of the solutions were calculated according to the regression equation, and the relative uncertainties were determined. The results are collected in Table 9.

## Precision for the determination of model rinse water samples

Three series of experiments were conducted. In the first series five model rinse water samples from injections with the concentration of α-GPC equal to 25 mg/l, in the second series five model rinse water samples from oral solution with the concentration of α-GPC equal to 12 mg/l, and in the third series five model rinse water sample solutions from

capsules with the concentration of α-GPC equal to 40 mg/l were prepared. The solutions were treated as described in the general procedure, and then the absorbance of the coloured product was recorded; the concentrations of the solutions were calculated according to the regression equation, and the relative standard deviations were determined. The results are collected in Table 10.

## Comparison of the results of titrimetric and spectrophotometric determinations

A total of 1.0000 g of α-GPC was weighted, dissolved in ca. 80 ml of glacial acetic acid, the solution was transferred to the 100 ml volumetric flask and the volume of the solution was adjusted by glacial acetic acid. Ten aliquots of 5.0 ml of the prepared solution were transferred to the titration flasks, 40 ml of acetic anhydride and 10 ml of the 3% solution of mercury (II) acetate was added to the each, the solutions were mixed and titrated with the standardised solution of 0.1 M perchloric acid using 0.05 ml of the 0.5% solution of crystal violet as indicator. Then ten aliquots of 5.0 ml of the prepared solution were taken, transferred to ten 2,000 ml volumetric flasks, and the volumes of the solutions were adjusted by water. The diluted solutions were treated as described in the general procedure, and then the absorbance of the coloured product was recorded, and the concentrations were calculated according to the regression equation. The content of α-GPC in the weighting was calculated using the both methods, and the F-test of equality of variances and the $t$-test of equality of means were performed. The results are shown in Table 11. As could be seen, both the calculated F- and t-values do not exceed the critical values for the given degrees of freedom and $p$-value, which means that these titrimetric and spectrophotometric methods give statistically equal results for the tested sample.

## DISCUSSION

The experiments show that the proposed spectrophotometric method is suitable for the determination of α-GPC both in pharmaceutical formulations and in industrial equipment cleaning rinse water. The method is simple; it does not require complicated sample preparation or sophisticated equipment. The method is selective with respect to the common excipients; however, different choline derivatives and other substrates and enzymes producing hydrogen peroxide might interfere, and the influence of more complex matrices was not studied. The molar attenuation coefficient equals 1,110 m$^2$/mol, the limit of detection equals 1 mg/l, and the limit of quantification equals 3.3 mg/l, the calibration curve is linear in the range from 1 to 40 mg/l of α-GPC with the good correlation coefficient, and the optimum range of α-GPC concentrations for determination is 5–16 mg/l. Therefore, this method is less sensitive than more comprehensive NMR, GC/MC and HPLC/MS methods, but more sensitive than HPLC/refractometric and CE/UV methods developed for determination of α-GPC in pharmaceutical formulations. The relative uncertainty for the analysis of pharmaceutical formulations does not exceed 2.5%, which is a fair value for the kinetic method; the relative uncertainty for the analysis of modelling industrial rinse water does not exceed 7%, which is also acceptable for cleaning validation sample analysis. The relative standard deviation does not exceed 2.5% for intra-, 3.5% for

Nikolaychuk (2023), *PeerJ Analytical Chemistry*, DOI 10.7717/peerj-achem.24

**Table 10** The precision test for the model rinse water sample solutions.

| Sample number | Experiment 1: Solution from intravenous injection, 25 mg/l | | Experiment 2: Solution from oral solution, 12 mg/l | | Experiment 3: Solution from capsules, 40 mg/l | |
|---|---|---|---|---|---|---|
| | Absorbance at 595 nm | Concentration of $\alpha$-GPC (mg/l) | Absorbance at 595 nm | Concentration of $\alpha$-GPC (mg/l) | Absorbance at 595 nm | Concentration of $\alpha$-GPC (mg/l) |
| 1 | 1.036 | 24.26 | 0.496 | 11.74 | 1.622 | 37.86 |
| 2 | 0.954 | 22.36 | 0.516 | 12.19 | 1.600 | 37.35 |
| 3 | 1.028 | 24.08 | 0.463 | 10.96 | 1.531 | 35.75 |
| 4 | 0.991 | 23.23 | 0.482 | 11.42 | 1.618 | 37.78 |
| 5 | 1.027 | 24.07 | 0.456 | 10.80 | 1.569 | 36.63 |
| Mean value | 1.007 | 23.59 | 0.482 | 11.42 | 1.588 | 37.07 |
| SD | 0.035 | 0.801 | 0.025 | 0.569 | 0.038 | 0.885 |
| RSD (%) | 3.43 | 3.39 | 5.08 | 4.98 | 2.40 | 2.39 |

**Table 11** The comparison of titrimetric and spectrophotometric methods.

| Sample number | Titrimetric method | | | Spectrophotometric method | | |
|---|---|---|---|---|---|---|
| | Volume of the titrant (ml) | Amount of $\alpha$-GPC in the aliquot (mg) | Amount of $\alpha$-GPC in the weighting (mg) | Absorbance at 595 nm | Concentration of $\alpha$-GPC in the aliquot (mg/l) | Amount of $\alpha$-GPC in the weighting (mg) |
| 1 | 1.96 | 50.42 | 1008.4 | 1,058 | 24,77 | 990,8 |
| 2 | 1.94 | 49.90 | 998.0 | 1,062 | 24,87 | 994,8 |
| 3 | 1.9 | 48.87 | 977.4 | 1,074 | 25,14 | 1005,6 |
| 4 | 1.94 | 49.90 | 998.0 | 1,066 | 24,98 | 999,2 |
| 5 | 1.92 | 49.39 | 987.8 | 1,072 | 25,11 | 1004,4 |
| 6 | 1.96 | 50.42 | 1008.4 | 1,058 | 24,78 | 991,2 |
| 7 | 1.92 | 49.39 | 987.8 | 1,064 | 24,93 | 997,2 |
| 8 | 1.92 | 49.39 | 987.8 | 1,059 | 24,80 | 992,0 |
| 9 | 1.92 | 49.39 | 987.8 | 1,061 | 24,84 | 993,6 |
| 10 | 1.94 | 49.90 | 998.0 | 1,075 | 25,17 | 1006,8 |
| Mean value | 1.93 | 49.70 | 994.0 | 1,065 | 24,94 | 997,6 |
| Sample variance | 0.0004 | 0.247 | 98.80 | 0.00004 | 0.0236 | 37.78 |
| F-value ($f_1 = 9, f_2 = 9, p = 95\%$) | | | 2.62 | Critical F-value ($f_1 = 9. f_2 = 9. p = 95\%$) | | 3.18 |
| t-value ($f = 18, p = 95\%$) | | | 0.92 | Critical $t$-value ($f = 18. p = 95\%$) | | 1.734 |

inter- day precision, and 5.5% for analysis of modelling industrial rinse water. The accuracy and the precision of the proposed method are comparable with those for other proposed methods and fall within the requirements for the analysis of industrial equipment cleaning validation samples. The analysis revealed no statistical difference between the proposed spectrophotometric method and the non-aqueous titration method of *State Pharmacopoeia of the Russian Federation (14th Edition) (2018)* in the determination of α-GPC in a bulk sample. The proposed method is not intended to compete with the sophisticated NMR, GC/MS and HPLC/MS methods for the analysis of α-GPC and other choline derivatives in complex matrices, but it presents a simple a quick solution when the quantification of residual amounts of α-GPC in an aqueous solution is needed. The method is recommended for the routine and quick analysis of α-GPC in industrial intermediate goods during the intermediate quality control, in pharmaceutical formulations and in industrial equipment cleaning rinse water.

## CONCLUSIONS

A simple spectrophotometric method for the determination of α-GPC in pharmaceutical formulations and industrial equipment cleaning rinse water using the enzyme glycerophosphocholine phosphodiesterase and the WAKO Phospholipids C assay kit was proposed. The method is based on the enzymatic hydrolysis of α-GPC to choline, the enzymatic oxidation of choline, the reaction of formed hydrogen peroxide with 3,5-dimethoxy-N-ethyl-N- (2-hydroxy-3-sulfopropyl)-sodium aniline and 4-aminoantipyrine, and the colourimetric determination of the formed product. The method shows good analytical performance, does not require lengthy sample preparation and sophisticated laboratory equipment, and is suitable for routine analysis.

### Funding
The author received no funding for this work.

### Competing Interests
Pavel Anatolyevich Nikolaychuk was employed by LLC "Velpharm" from 10.02.2020 until 30.06.2021.

### Author Contributions
- Pavel Anatolyevich Nikolaychuk conceived and designed the experiments, performed the experiments, analyzed the data, prepared figures and/or tables, authored or reviewed drafts of the article, and approved the final draft.

### Data Availability
   The raw data is available in the Supplemental Files.

## Supplemental Information

Supplemental information for this article can be found online at http://dx.doi.org/10.7717/peerj-achem.24#supplemental-information.

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
