# Peer review of "Spectrophotometric determination of L-α-glycerylphosphorylcholine in pharmaceutical formulations and industrial equipment cleaning rinse water with the WAKO Phospholipids C assay kit"

_PeerJ Analytical Chemistry, doi:10.7717/peerj-achem.24_

## Round 0.1 · original submission · Major Revisions

Thank you for submitting your work to PeerJ Analytical Chemistry. In your revised manuscript, please address the individual and detailed comments as well as suggestions of the reviewers. Also, consider making plots of some of the tabulated data with statistical information annotated in the description. This would present a more visual way to understand the trends in your analyses. We look forward to reviewing your revised manuscript.

·

Basic reporting

- Introduction fails to demonstrate WHY this method is relevant. While I think it is a nice concept, it’s usefulness is questionable. I would not consider NMR or mass spectrometry outside the possibility for most organizations, even most public institutions have access to these technologies. Further, I’m unclear as to why being able to quantify GPC in waste products actually matters? It isn’t an environmental hazard? As for OTC-nature of GPC, aside from a teaching lab, I don’t see the relevance. I’m not saying it *isn’t* relevant. I’m saying the point is not clearly stated as to why anyone would want to use this method.
- The results section feels more like a methods section rather than actually explaining the results obtained. For example in “Comparison of the Results of Titrimetric and spectrophotometric determinations” you describe how you did it but not what the results actually mean. I would like to see a deeper explanation of results.
- Specific instances below but I’ve found many statements to be under-cited in the introduction.
- In some instances (example line 71-72), the citation seems to be part of the sentence but the use of parenthesis is inappropriate.
- Due to the analytical nature of this paper, I would like to see some information about the calibration of the spectrophotometer.
- Usually a headquarters/location is reported for companies
- The figure legends are lacking information. They are extremely brief and could use more information.
- Table 3: I think the absorbance values should have a “.” Instead of a “,”.
- Table 6 is very difficult to read. I recommend abbreviations if possible.
- The last sentence needs to be broken up into multiple
- Introduction:
- Line 44-46: Are the first and second point related? By what metric is GPC performance enhancing? Also, changes in GPC is definitely a hallmark of cancer and there are many more newer sources supporting this.
- Lines 47-49: I would like to see citations for both these statements.
- Line 57-60: This statement is under cited. I would like to see citations for each method.

Experimental design

-By far the biggest problem is that the source/number of experimental GPC-containing samples is not reported. Is there one formulation for each (oral, intravenous, and capsule) or multiple? Are the multiple samples mixed together from the same package/lot or different ones? Are the formulations the same or are there other components which may affect the output of the assay? Were the reported concentrations of GPC validated?
-Table 2: I am a little concerned that they did not continue this experiment until the color change stopped. At some point you should hit a maximum. The text indicates this is the “optimal absorbance” but how is that actually decided?
-Model Waste Solutions: I’m curious as to why a model is being used rather than actual waste water? I GPC in waste water is truly of concern, I would like to see real world samples tested.
-Line138-139: This statement is not sufficient. What local market? Who made the formulations? How many were procured and tested?
-Line 151/Line 148: How long was it stored for and how long can it be stored for? Also, what temperature is the refrigerator?
-Line 181: Where working solutions kept at room temperature? How long were they kept before experimentation? The half life of GPC is less than a working day so this is important information.
-Line 182-188: How was the stock determined to be 250 g/L? Is that what was reported on the packaging from the company? Or was it measured? If it was not measured, I would highly recommend a secondary method of quantitation. Further, what is the age of the GPC ampules and prior storage? Was degradation prior to experimentation taken into account?
-Line 189-105: Same as above.
-Line 210/218/226: How are you arriving at these numbers? Is there any secondary validation?

Validity of the findings

-Discussion is quite lacking. It is not putting this assay into any sort of context that makes it compelling.
-It is repeatedly mentioned that this is a “quick” analysis but there is very little discussion of the actual amount of time beyond how long the samples are heated for. I would like to see some commentary on how long this method takes compared to both the titration method as well as other methods such as MS or NMR.

Additional comments

I think the idea of this paper is quite interesting and I love the idea of taking a commerical assay kit for another purpose, especially exploiting an added enzyme. I just feel like this paper lacks context and a real world application. I would also like to see more in terms of comparisons to other ways of measuring GPC. I understand the author may not have access to something like NMR or MS but even a theoretical comparison would be interesting (ie what is the detection limit of your method compared to NMR for example). I would like to see experiments but also understand that there may be limitations in this case.

Reviewer 2 ·

Basic reporting

Writing of this manuscript has left a lot to be desired. Various small errors, typos leads to confusion and misunderstanding. The article does not present a reasonable scientific writing level to be refereed with.

Literature reference and expression was quite confused sometimes. The way of citing literature in manuscript needs to be refined. Reference list is confusing at some places.

No explanations or speculations were given as to why the EMEA and FDA lacks regulatory monograph on the residual alpha-GPC in the industrial environment and production wastes. No revelation on the reasons behind the positive regulatory needs for alpha-GPC in the industrial environment and product wastes by Russian Federation. The lack of which significantly attenuates the significance, and may likely nullify the justification of this study.

The author only provided the detection results from the "simulated wastes". Lack of actual quantitation data of alpha-GPC in actual wastes does not corroborate with the initial goal set forth in the study. If the assay is so easy to use, why no actual detection and evaluation was performed?

Experimental design

No matrix effect was discussed. This renders the precision and accuracy in the detection of alpha-GPC in actual waste content by this colorimetric spectrometric method unknown.

Validity of the findings

The results basically were just the outcome of using Wako phospholipids C kit on detecting the alpha-GPC in well-defined substance. I don't see the scientific merits of this outcome.

When some of the results read like method, it is hard to comment on anything.

Additional comments

Quite a lot of the supplementary information provided was not documented in English, Such information belongs to the auditing documentation for the regulatory body of local or state government, which is not necessarily required by scientific journals. This does not facilitate the understanding of the content of the manuscript.

Annotated reviews are not available for download in order to protect the identity of reviewers who chose to remain anonymous.

---

## Round 0.2 · Major Revisions

Many issues have been addressed by the authors. Unfortunately a few more issues have been identified, and these need to be addressed as well.

Reviewer 3 ·

Basic reporting

No comment

Experimental design

The ability to measure GPC with a commercial kit is within the expected range given the measurement method of this kit, and we do not believe that we have obtained any new findings.
This is an interesting example of a commercial kit application, but I do not think it is versatile on its own.
For example, we would like to see the method verified as having broad versatility, such as being able to measure GPC in blood or GPC in food.

Validity of the findings

GPC can be easily measured by LC-MS/MS; however, since LC-MS/MS is an expensive analytical instrument, it may not be available to everyone. Therefore, the usefulness of this simple kit is understandable. However, it is necessary to indicate whether the required measurement accuracy for the measurement of various rinse solutions used in the manufacturing of this formulation can be met with concentrations within the measurement range of the kit.

Reviewer 4 ·

Basic reporting

The MS described a lot of information, including the lengthy description on the parts of introduction and sample preparation, however, without major focus related with the study design. Furthermore, the key point of the reaction mechanism between the WAKO Phospholipids C assay kit and the target analyte is not clear, so it is hard to evaluate the study in the perspective of science and innovation.

Experimental design

The research question is not clear, so the experimental design seems in a mess.

Validity of the findings

It is quite hard to find the novelty of this study.

Additional comments

It is desired to supply meaningful description on its major question-driven discussions.

---

## Round 0.3 · accepted · Accept

Given the level of positive response from the reviewer, I am now pleased to accept the paper for publication in PeerJ Analytical Chemistry.

·

Basic reporting

Basic reporting is appropriate.

Experimental design

Experimental design is appropriate for the journal and well reported.

Validity of the findings

The data is robust and beliable. Conclusions are appropriate for the article.

Additional comments

This is certainly not the highest impact work but I think it is acceptable in it's current state.